# An Attempt to Track Two Grades of Road Bitumen from Different Plants Using Fourier Transform Infrared Spectroscopy

**DOI:** 10.3390/ma14195870

**Published:** 2021-10-07

**Authors:** Serge-Bertrand Adiko, Alexey A. Gureev, Olga N. Voytenko, Alexey V. Korotkov

**Affiliations:** 1Department of Oil Refining Technologies, National University of Oil and Gas “Gubkin University”, 119991 Moscow, Russia; a.gureev@mail.ru; 2Research Centre of Gazpromneft-Bituminous Materials, 390047 Ryazan, Russia; voytenko.on@gazprom-neft.ru (O.N.V.); korotkov.av@gazprom-neft.ru (A.V.K.); 3Scientifical and Educational Centre Bituminous Materials, National University of Oil and Gas “Gubkin University”, 119991 Moscow, Russia

**Keywords:** Fourier Transform Infrared Spectroscopy (FTIR), spectrometry indices, road bitumen, ageing, variance coefficient, tracking

## Abstract

This study aimed to evaluate the possibility of using Fourier Transform Infrared (FTIR) spectroscopy to track binders produced by three different plants: plants A, B, and C. The work included the quality assessment of 80 bituminous materials graded as BND 70/100 and 100/130 according to GOST 33133 (Russian interstate standard) and chemical analyses using FTIR spectroscopy. FTIR analyses were conducted before and after short-term ageing in a Rolling Thin Film Oven Test (RTFOT). Thus, the number of binder samples was multiplied by two (2) for a final total of 160 infrared (IR) spectra. All infrared spectra were normalised to ensure the reliability of results, and the standard deviation and variance coefficient were included. The principal purpose of the present work was to track the origin and the ageing extent of the bituminous binders under study.

## 1. Introduction 

The traditional type of organic binder is petroleum bitumen or petroleum bituminous materials [1]. Bituminous materials are one of the most important tonnages produced by the petroleum refining industries, as bituminous materials are also one of the most widely used materials in the world [2,3,4]. The widespread use of bituminous materials in many fields of human activity, such as insulating and binding materials for road construction, airfield repair, pavements, roofs, etc., is mainly due to their physicochemical characteristics [5]. Improving or optimising the physicochemical characteristics of bituminous materials is a challenge for the scientific community today in order to meet the modern requirements of our society or our civilization. However, bitumen production technology differs from region to region depending on ecological, climatic, and sanitary conditions or requirements [2]. As a result, the technical specifications for the quality of bituminous materials produced in leading countries such as the United States of America, China, Canada, Spain, the Russian Federation, and France differ. Climatic conditions, road traffic density, and the technology used to produce bituminous materials contribute to these differences [5,6,7].

In the Russian Federation, oxidised bitumen is used as road bitumen as prescribed by the GOST 33133 standard. The most common binders’ grades of GOST 33133 are BND 100/130, 70/100, and 50/70 [5]. The road bitumen in this region of the world is subject to higher requirements in terms of frost resistance (Fraass breaking point) and high-temperature resistance (softening temperature), as well as in terms of adhesive properties [8].

The process of bitumen manufacture from crude oil is rather complex [3,4]. Oxidation, cracking (vacuum distillation with rectification), and compounding are fundamental techniques used to produce petroleum bituminous materials [3,4,5,9]. The primary raw material used for the production of petroleum bitumen is tar or vacuum residue. The technology of manufacturing bitumen involves several stages, regardless of the production technique, starting with the desalination of the crude oil, then atmospheric and vacuum distillation [3,10,11]. The most widely used bitumen manufacturing process in the Russian Federation is the air blowing process due to its simplicity, reliability, and cost-effectiveness. Furthermore, Russian legislation allows air-blown or oxidised bitumen as a binder for pavement or roadways, which is not the case in many Western countries due to their laws and technical requirements [5]. However, other processes are also used in Russia, such as compounding to manufacture polymer-modified bitumen [3,12].

Most specifications and standards are based on the physicochemical properties of bituminous binders, measured empirically [5,6,13]. Thus, for many manufacturers and even dealers, one of the crucial objectives to ensure a good reputation with these customers is to guarantee the physicochemical properties of the binders. This comprehensive guarantee implies the possibility of identifying and tracking its products among other products in the same product categories. Identification is therefore one of the important goals of mass production. Gazpromneft-Bituminous Materials, in this approach, undertakes to develop traceability methods for its manufactured products. One of the most suitable identification methods is to study the product’s chemical structure to determine its distinctions and thus solve the traceability challenges.

Moreover, economically, the identification or tracking method chosen or developed should not be expensive, but fast and efficient. Here, FTIR analysis was positioned as the method of the first choice for the study of road bitumen or asphalt pavements [14,15]. The choice was based on FTIR analyses’ reputation in the field of bituminous materials, as evidenced via numerous published studies, such as those of the Western Research Institute and Petersen [16]. As already mentioned, this method of experimental analysis is frequently used because it provides reliable and precise information on the chemical structure of the samples studied [15,16].

The chemical structure of bitumen or bituminous materials is complex. Four chemical fractions are obtained using the bitumen separation method ASTM D 4124-01 or ASTM 2007 [9,15,16]: asphaltenes, resins, aromatic compounds, and saturated compounds. Bitumen, in its structure, is a colloidal system in which asphaltenes or asphaltene micelles are dispersed in maltenes. The middle of the dispersion of maltenes includes resins and oils (saturates and aromatics) [10]. Resins, i.e., the polar components of maltenes, stabilise the micelles of asphaltenes.

Consequently, the oxidation or ageing process includes many oxidative reactions that cause the generation of transitions from one polarity-based fraction to another [2,3,10]. The ageing of bituminous materials is an important problem because sulphoxides and carbonyl groups are formed during the oxidative ageing process [16,17,18]. Carbonyl groups in bitumen include carboxylic acid, ester, ketone, aldehyde, anhydride, and amide [14,17]. Therefore, many studies have focused on anti-ageing to find lasting solutions for thermochemical stability [14,16]. One of the modern solutions is the introduction of epoxy-based compounds as anti-ageing technologies for bituminous materials [15,19,20].

Following the literature, FTIR analysis is one of the most effective analytical methods for studying the chemical composition of oil, vacuum residues, and bitumen [21]. The range from 4000 cm^−1^ to 400 cm^−1^ is the interval where the fundamental molecular vibrations of functional groups are displayed in the spectrum as absorption bands [14,15,22,23]. According to field studies, bands at 2920 and 2850 cm^−1^ are assigned, respectively, C–H aliphatic asymmetric and symmetric stretching vibrations [14,17,21]. The absorption region of 1740–1710 cm^−1^ corresponds to the C=O stretching of carbonyl groups. It is good to report that the band at 1710 cm^−1^ often determines the oxidation degree of oils and bitumen. The band at 1600 cm^−1^ is associated with C=C due to its aromatic compounds, and 1030 cm^−1^ the bands’ sulphoxide groups. The bands at 1375 and 1455 cm^−1^ are assigned to the CH_2_ groups > 4, CH and CH_2_CH_3_ groups. The paraffin structures are especially at the band at 720 cm^−1^ [5,22,24]. Under 1500 cm^−1^, known as the fingerprinting region [25,26], is a complex and informative part of the IR spectrum that typifies the molecule under study. Especially, in our particular case, this is the bitumen.

The binder tracking is topical because oxidised bitumen is mainly used for roofs, floor coverings, waterproofing complexes, adhesives, and anticorrosion products according to standard EN 13304 [5,13]. As mentioned above, in the Russian Federation, oxidised bitumen is used as road bitumen and as a feedstock for modified bitumen and other bituminous materials [5]. Road projects are frequently supplied with various bituminous materials from various factories, manufacturers, and dealers [27,28]. During road project operations, bituminous material passports (origin) can get lost. In this regard, there are cases where customers (road projects) are not satisfied with the quality of the products supplied [29]. Thus, such cases can become a source of conflict between dealers, manufacturers, and customers. Maintaining good business relationships with customers or dealers is essential for any manufacturer. One solution is to identify or track bituminous materials by manufacturer, dealer, or customer, which helps maintain the partnership. Furthermore, many oil plants around the world use almost identical sources of oil for their production. For example, in Russia, crude oil supplied to refineries is usually delivered via pipelines [30,31], thereby having the same crude oil.

The investigation aimed to discover or find the differences between bituminous binders from different plants using FTIR in order to recognise or track their origin. FTIR analyses were performed before and after short-term ageing in the Rolling Thin Film Oven Test (RTFOT), as shown in Figure 1. The study also included the physicochemical properties assessment of bituminous binders graded as BND 70/100 and 100/130 according to GOST 33133.

## 2. Materials and Methods 

### 2.1. Materials

The materials in this study were 80 samples of road bitumen (BND) graded as BND 70/100 and 100/130, according to GOST 33133, collected directly under GOST 32268-2013 (standard for sampling bituminous materials) from three Gazpromneft-Bituminous Materials plants (plants installations) over six months. Binders were sampled at 40 specimens BND 70/100 and 40 specimens BND 100/130, as shown in Figure 2. Note that the survey was carried out before and after short-term ageing (RTFOT under GOST 33140) and that all the samples studied were formulated by the air-blowing process from different batches, that is, different production periods.

### 2.2. Methods

The advancement of technologies and methods offers a multitude of possibilities for studying bitumen and, in particular, its chemical structure. In Table 1, the Russian standard GOST 33133-2014 was used in addition to the FTIR [8]. As well, the variance coefficient (CV) was applied due to the size of the data, and it was defined as the standard deviation divided by the mean. The algorithm for the variance coefficient and standard deviation, see Equations (1) and (2) below, makes it possible to measure the dispersion of the data obtained and facilitate their analysis [32].
(1)$$\mathrm{Variance}\;\mathrm{coefficient}\;(\mathrm{CV})=\frac{SD}{|M|\;}\times 100$$

*SD* is the standard deviation, while *M* is the mean.
(2)$$\mathrm{Standard}\;\mathrm{deviation}\;(\mathrm{SD})=\frac{\sum |X-M|}{N\;}$$
where *N* is the number of binder samples by plants, *X* is each value of variables, and *M* is the mean.

It was deliberately decided that the variance coefficient be set at 30% due to certain factors, such as the production batch and the origin of the binders.

#### 2.2.1. Russian Interstate Standards for Road Bitumen: GOST 33133–2014

#### 2.2.2. Infrared Spectroscopy Analysis: ATR-FTIR Spectroscopy

The study was conducted before and after short-term ageing (RTFOT under GOST 33140). IR spectra of samples in the range of 4000–400 cm^−1^ (fundamental region) were obtained using the IR Affinity-1S spectrophotometer (Shimadzu, Kyoto, Japan) with the console Quest ATR Diamond (Single-Reflection) at room temperature between 22 and 27 °C. The binder samples were placed on a diamond surface, and interferograms were recorded with a 4 cm^−1^ resolution and averaged over 30 scans. After subtracting the baseline, the obtained IR spectra were normalised to the absorption band of 2920 (±3) cm^−1^ according to Equation (3): (3)$${\mathrm{Abs}}_{\mathrm{norm}}\;(\mathrm{wn})=\mathrm{Abs}\;\left(\mathrm{wn}\right)\times \frac{1}{\mathrm{Abs}(2920\;\left(\pm 3\right){\;\mathrm{cm}}^{-1})}$$
where Abs_norm_ (wn) is the absorbance spectrum normalised as a function of wavenumber; Abs (wn) is the original absorbance spectrum; Abs (2920 (±3) cm^−1^) are original absorbance values at 2920 (±3) cm^−1^. The normalisation of this band resulted in an absorbance value set at 1.0, and the full spectrum was multiplied by a ratio factor, as shown in Figure 3 [33]. The IR spectra’ normalisation, as fixed in Hofko, B., Alavi, M.Z. et al., was based on the idea to eliminate any changes in the absorbance spectra due to the variation of the infrared beam penetration between samples, which would bias the interpretation of results [33]. The binder samples’ chemical structures via IR were based on the calculation of spectrometry indices defined as the ratio of the absorbance (a.u.) values in the maxima of the corresponding absorption bands in Equations (4)–(9) [5,34].
(4)$$\mathrm{Aromaticity}\;\mathrm{indices}=\frac{1600{\;\mathrm{cm}}^{-1}}{720{\;\mathrm{cm}}^{-1}}$$

Aromaticity indices are aromatic rings that make up the structure of bitumen to the relative content of aliphatic fragments.
(5)$$\mathrm{Oxidation}\;\mathrm{or}\;\mathrm{carbonyl}\;\mathrm{indices}=\frac{1710{\;\mathrm{cm}}^{-1}}{1465{\;\mathrm{cm}}^{-1}}$$

Oxidation or carbonyl indices (oxidation rate) are parameters that characterise the oxidation degree. An increase in the oxidation degree accompanies the oxidative ageing process. In other words, this is the increase in oxidised compounds such as carbonyl groups in the bitumen.
(6)$$\mathrm{Isomerisation}\;\mathrm{indices}=\frac{1380{\;\mathrm{cm}}^{-1}}{1465{\;\mathrm{cm}}^{-1}}$$

Isomerisation indices describe the branched-chain compound hydrocarbons in the chemical structure of bitumen.
(7)$$\mathrm{Aliphaticity}\;\mathrm{indices}=\frac{720{\;\mathrm{cm}}^{-1}+1380{\;\mathrm{cm}}^{-1}}{1465{\;\mathrm{cm}}^{-1}}$$

Aliphaticity indices are aliphatic fragments to the relative content of aromatic compounds present in bitumen samples.
(8)$$\mathrm{Sulphurisation}\;\mathrm{indices}=\frac{1030{\;\mathrm{cm}}^{-1}}{1465{\;\mathrm{cm}}^{-1}}$$

Sulphurisation indices represent the sulphurised compounds in the chemical structure of bitumen.
(9)$$\mathrm{Asphaltenes}\;\mathrm{indices}=\frac{880{\;\mathrm{cm}}^{-1}}{820{\;\mathrm{cm}}^{-1}+750{\;\mathrm{cm}}^{-1}}$$

Asphaltenes indices are the polycondensed aromatic compounds in the chemical and colloidal structure of bitumen.

Based on the set goals and origin of bituminous binders, the variance coefficient of chemical structures was set at 30%.

## 3. Results and Discussions

### 3.1. Physicochemical Properties of Binder Samples by Plants

The samples’ physicochemical qualities under GOST 33133 for grades BND 70/100 and 100/130 are the focus of the first portion of the inquiry. The physicochemical properties of binder samples from manufacturers or plants are presented in Table 2, Table 3, Table 4 and Table 5.

During the evaluation of the samples, it was discovered that nearly half of the BND 70/100 binder samples from plant A exhibited insufficient “ductility at 0 °C,” i.e., nine (9) of the twenty (20) samples. Of the nine (9) samples that did not meet the “ductility at 0 °C” criteria, five (5) developed a rupture (crack) since the start of the tests. In brief, 45% of BND 70/100 from plant A were not BND 70/100 after being evaluated according to GOST 33133 standards. 

During the study, the BPA 5 breakpoint analyser was damaged. Some samples were not tested. As a result, when compared with other indicators, the statistical data for the Fraass breaking point can be regarded as insufficient.

In the first part of our research, we decided to establish the means of physicochemical properties to find some differences between the BND presented by plants and to evaluate the physicochemical properties of bituminous binders under the standard GOST 33133. The variance coefficient must be less than or equal to 30% for a statistic to be valid. This value of 30% is motivated by the fact that there are three (3) fundamental factors in the production of bitumen by air blowing: temperature, pressure, and airflow rate/raw material. We considered 10% as the variance coefficient due to corrections based on technical problems during the production process for each factor. Thus, the sum of the variance coefficient of each factor is equal to 30%, which is the variance coefficient of the production process on the physicochemical properties and chemical structures by the bituminous binders’ species. Moreover, the four bituminous binders or four bituminous binders species studied were manufactured somewhat differently, and each binder sample came from different production batches. The bituminous binders are graded into pairs as BND 70/100 and 100/130 according to GOST 33133. Therefore, they must have very close physicochemical properties in pairs. The intermediate results with the remarks are presented in Table 6 below:Most of the studied samples had physicochemical properties under GOST 33133 requirements for BNDs 70/100 and 100/130. However, it should be noted that the samples studied here are far from being superior bituminous binders due to the low margin recorded.The test results further showed that 45% of the BND 70/100 samples and 60% of the BND 100/130 samples from Plant A did not meet GOST requirements, including “ductility at 0 °C” for BND 70/100 and “mass loss (%) after RTFOT ageing” for BND 100/130. Regarding “ductility at 0 °C” for BND 70/100 from Plant A, five (5) showed failure (crack) since the testing began. Thus, these five (5) measurements were not included in the statistical mean. On the other hand, concerning the property “mass loss (%) after RTFOT ageing” of BND100/130 from plant A, the parameters of the standard deviation/variance coefficient confirmed the remark. However, the arithmetic means of “mass loss (%) after ageing RTFOT” ET 100/130 from plant A was following GOST 33133, as for BAND 70/100 from plant B.Regarding the identification or tracking of bituminous binders, it is noted that the “rotational viscosity at 1.5 s^−1^ and 60 °C, Pa∙s before RTFOT” of the bituminous binder samples from plants A and C was generally less than 200 Pa∙s before RTFOT, and after short-term ageing, less than 500 Pa∙s. On the other hand, the rotational viscosity values of bituminous binders from plant B were close to 400 Pa∙s before RTFOT and 900 Pa∙s after RTFOT.

### 3.2. Results of Fourier-Transform Infrared Spectroscopy Investigation

Studies using IR spectroscopy are divided into two parts: visual analysis of IR spectra and calculation of spectrometry indices with the interpretation of the data obtained. Visual analysis of IR spectra was performed to find and determine functional groups. A visual analysis should establish whether or not there are differences between one sample and another for chemical compounds.

Each binder sample in the infrared spectroscopy investigation was studied before and after short-term ageing (RTFOT). Thus, the initial number of binder samples studied was multiplied by two (2) to study the samples after short-term ageing. Therefore, 160 samples were studied, which is equivalent to 160 infrared spectra with their spectral data. 160 IR spectra (samples) were shared between four species of binders or types of binder. For each type of bituminous binder, there were 40 IR spectra distributed as follows: 20 IR spectra before and after short-term ageing.

In order to measure and ensure the reliability of the results obtained, we also included the standard deviation and the variance coefficient in this study, while maintaining the same standard of statistical validity at 30%. FTIR analyses of bituminous binders or binder samples by plants yielded the following IR spectra shown in Figure 4, Figure 5, Figure 6, Figure 7, Figure 8 and Figure 9. Figure 4 and Figure 5 show 80 IR spectra of BND 70/100 from plants A and B, distributed as follows: 40 IR spectra before and 40 IR spectra after short-term ageing (RTFOT). The same distribution of IR spectra for BND 100/130 from plants A and C apply to Figure 6 and Figure 7.

The enumeration and detection of each band or number of waves of chemical compounds of the binder samples studied are shown in Table 7. Likewise, the results of spectrometry indices data are shown in Table 8, Table 9, Table 10 and Table 11. Using spectrometry indices to analyse the chemical structure of bituminous binders can help detect, track, and understand the stability of their chemical composition.

As shown in the figures above, visual inspection of the IR spectra did not yield the expected results. Indeed, all the samples of binder studied had almost identical chemical functions, as can be seen in Table 7. Except for BND 70/100 binders from Plant A, we found two peaks at 1710 cm^−1^ and 1600 cm^−1^ in virtually all other IR spectra of binder samples, precisely in the range 1590–1740 cm^−1^. The peculiarity of the BND 70/100 from plant A compared with the others studied was that they only had the band at 1600 cm^−1^. This observation is supported by the IR spectra of BND 70/100 from plant A before and after RTFOT shown in Figure 4, Figure 5, Figure 8 and Figure 9. All BND 70/100 and 100/130 IR spectra from plants B and C had bands at 1710 and 1600 cm^−1^, as well as BND 100/130 from plant A.

Briefly, the only visual difference was seen in the range of 1740–1690 cm^−1^ (1710 cm^−1^), while the carbonyl, hydroxyl, and ether (C=O) groups were among the chemical classes represented by the 1710 cm^−1^ band. According to the spectral inspection results of BND 70/100 from plant A, these BND 70/100 are produced without oxidation or with minimal oxidation, that is, under mild oxidation conditions. This interpretation of the BND 70/100 binder samples is due to their IR spectra, in which there is virtually no peak at 1710 cm^−1^.

Further analysis was performed to identify or track, including quality control of the binder samples, for the reasons stated above. This in-depth or further analysis was based on the oxidation degree via oxidation indices (spectrometry indices). The spectrometry indices were defined as the ratio of the absorbance values (a.u) to the maxima of the corresponding absorption bands. Before calculating the spectrometry indices, the baselines of the IR spectra were subtracted and the IR spectra were normalised.

The chemical structures of the analysed binder samples are or should be reflected by the spectrometry indices. The algorithm for calculating spectrometry indices includes the normalisation. The normalisation was carried out to eliminate any modification of the absorbance spectra, due to a variation in the penetration of the infrared beam between the samples which would distort the interpretation of the results. Among the spectrometry indices, one of the most important is the oxidation indices due to the visual differences observed in the IR spectra of the binder samples. The oxidation indices, or oxidation rate, are the parameters which characterise the degree of oxidation. The increase in carbonyl compounds characterises the process of chemical oxidation of oils and bitumen, as already mentioned in the introduction. The sulphurisation and aromatisation processes also accompany the chemical oxidation process. Therefore, other indices, such as aromaticity and sulphurisation, were included in our studies to confirm the results on the degree of oxidation.

The BND 70/100 spectrometry indices of plant A corresponded to the observations mentioned above. In this sense, the oxidation indices of BND 70/100 from plant A were the lowest because there are no bands of carbonyl compounds in their IR spectra. Furthermore, during our visual inspection of the infrared spectra samples, another phenomenon caught our attention. We observed that the IR spectra of the 100/130 BAND of plant A had two bands at 1710 and 1600 cm^−1^ in the range of 1740–1590 cm^−1^. Visual evaluation of the IR BND 100/130 spectra of plant A was confirmed by spectrometry indices (see Figure 6, Figure 7, Figure 8 and Figure 9). The degree of oxidation of BND 100/130 from plant A, on the other hand, was significantly higher than the BND 70/100 of the same plant, according to spectrometry indices. This fact, from our point of view, is somewhat peculiar. BND 70/100 and 100/130 are derived from the same plant A. If BND 70/100 and 100/130 are made with the same technology, BND 100/130 should have an oxidation grade of less than 70/100, not the other way around, in the sense that the 70/100 performance band is harder than BND 100/130. Based on our observations, there is every reason to believe that the production technology used for BND 70/100 differs from the technology used for BND 100/130. In this case, this assumption is the only rational one. It is perhaps conceivable that the raw materials for BND 70/100 and 100/130 are, though unlikely, implausible because Plant A is supplied via pipeline. Nevertheless, Table 8 and Table 9 show that the BND 70/100 and 100/130 binder samples follow the ageing law, because the coefficient of variance after short-term ageing (RTFOT) was within the norm (standard); see Summary Table 12.

Looking at the IR spectra (Figure 4, Figure 5, Figure 8, and Figure 9) and spectrometry indices (Table 10) of BND 70/100 from plant B, we observed a significant (high) oxidation degree before and after short-term ageing. According to Table 10 and Table 12, the degree of oxidation before and after short-term ageing (RFTOT) is plus or minus 0.160, which shows that the chemical structure of these binders is generally stable, as shown in Figure 10. The regular variance coefficient and the standard deviations confirm these observations. The values of the other spectrometry indices in Table 12 confirm this assumption.

The analysis of BND 100/130 from Plant C seemed to be the most curious of all the results of the present study. Curiosity was found in the results obtained. The variance coefficient before ageing (RTFOT) for the oxidation indices was 51.2%, as shown in Table 12, and therefore outside the standards set. This allows us to conclude, at this stage, by the dispersion of the data, that the BNDs from plant C are not very stable. In addition, post-ageing studies (RTFOT) showed that the values of the spectrometry indices change in the opposite direction of ageing. Usually, sulphoxides, carbonyl groups and aromatic compounds increase upon oxidation of the binder, as shown earlier in the introduction. Therefore, after oxidation, an increase in the content of sulphoxides, carbonyl groups, and aromatic compounds is to be expected. However, for the BND from plant C, the mean values of the spectrometry indices showed the opposite of the results expected after short-term ageing (RTFOT). The oxidation indices of BND 100/130 binder samples before RTFOT were higher than after short-term ageing (RTFOT). On the other hand, the proportion of sulphoxides decreased, as did carbonyl groups and aromatics. Rejuvenation of the binder samples (chemical structure) was logically observed. The rejuvenation phenomenon is a positive process because it guarantees the quality of the binders during operation life. Unfortunately, as shown in Table 11, all BND 100/130 binder samples after short-term ageing (RTFOT) did not follow the rejuvenation phenomenon because some samples showed a high degree of oxidation. Therefore, rejuvenation may be an identifying or tracking factor for the BND 100/130 binder samples from Plant C.

Over the study of bituminous materials, it was noted that there are certain correlations between the FTIR spectroscopy data and the physicochemical properties in Table 2, Table 3, Table 4, Table 5 and Table 6. More precisely, the correlation is significant between the chemical structures and rheological data. For example, according to FTIR analyses, the bituminous binders from plant C exhibited the relatively highest and most stable degree of oxidation, accompanied by fairly high asphaltene, aromaticity, and sulphurisation indices, which are reflected in the highest values for “rotational viscosity at 1.5 s^−1^ and 60 °C” before and after short-term ageing (RTFOT). In addition, the bituminous binders from plant C with low aliphaticity indices, probably low in n-paraffin, had good physicochemical properties at zero and negative temperatures. On the other side, t BND 70/100 bituminous binders from plant A had poor physicochemical properties at zero or low temperatures (see ductility at 0 °C) with very high aliphaticity indices. At the same time, the BND 70/100 bituminous binders from plant A also showed relatively low oxidation indices before and after short-term ageing (RTFOT) according to FTIR results, which is visible in the values of “rotational viscosity at 1.5 s^−1^ and 60 °C.

## 4. Conclusions

The binder tracking is topical as road projects are often supplied with various bituminous materials from different manufacturers and dealers. During road project operations, bituminous material passports (origin) can get lost. In this regard, there are cases when customers (road projects) are not satisfied with the quality of the product supplied. Many oil plants around the world have almost the same sources of oil. For example, in Russia, crude oil supplied to refineries is usually carried out via pipeline. Therefore, this investigation was conducted to find the differences between bituminous binders from different plant facilities using FTIR analyses to track their origin. The work also included the assessment of the quality of bituminous binders graded as BND 70/100 and 100/130 according to GOST 33133.

As a result of the first part of the work, the road bitumen was tested according to GOST 33133, as shown in Table 13. Most of the bituminous binders studied have physicochemical properties under the requirements of GOST 33133 for TAPES 70/100 and 100/130, which are consistent with the standard deviation/variance coefficient. However, the test results obtained showed that 45% of the BND 70/100 samples and 60% of the BND 100/130 samples from Plant A did not meet GOST requirements, including “ductility at 0 °C” for BND 70/100 and “mass loss (%) after RTFOT ageing” for BND 100/130. Regarding the identification or tracking of bituminous binders, it is noted that the “rotational viscosity at 1.5 s^−1^ and 60 °C, Pa∙s before RTFOT” of the bituminous binder samples from plants A and C was generally less than 200 Pa∙s before RTFOT and less than 500 Pa∙s after short-term ageing. On the other hand, the rotational viscosity of the bituminous binders from plant B was close to 400 Pa∙s before RTFOT and 900 Pa∙s after RTFOT.

According to the results of the second part of the investigation (FTIR analyses), it can be concluded that the chemical structures of the binder samples before and after short-term ageing (RTFOT) were mostly stable. Most binder samples obeyed the law of ageing, except for BND 100/130 from Plant C, which was unpredictable, i.e., complicated statistical series. Therefore, the spectral data and spectrometry indices of the binder samples provided information about their differences; see summary Table 14 of experimental studies on FTIR spectroscopy.

In summary of the investigation, there were some differences between NDBs from different plants. The significant differences in Table 14 can be used to partially identify the bitumen, i.e., recognise or track its place of manufacture (origin). However, these differences can only be used in Gazpromneft-Bituminous Materials. The explanation lies in the fact that the chemical structures of BND 70/100 and 100/130 from competitors (other companies or plants) are unknown. Therefore, this point shows the limitations of this tracking method to identify or recognise products made by Gazpromneft-bituminous materials. Consequently, it is recommended to develop and introduce unique chemical markers (tracers) to achieve effective traceability and identify Gazpromneft-Bituminous Materials products.

On the other hand, this work can be considered as a start for new research using FTIR spectroscopy as an internal traceability tool for large companies such as Gazpromneft-Bituminous Materials or to explore other experimental methods more suited to effective traceability of bituminous binders.

## Figures and Tables

**Figure 1 materials-14-05870-f001:**
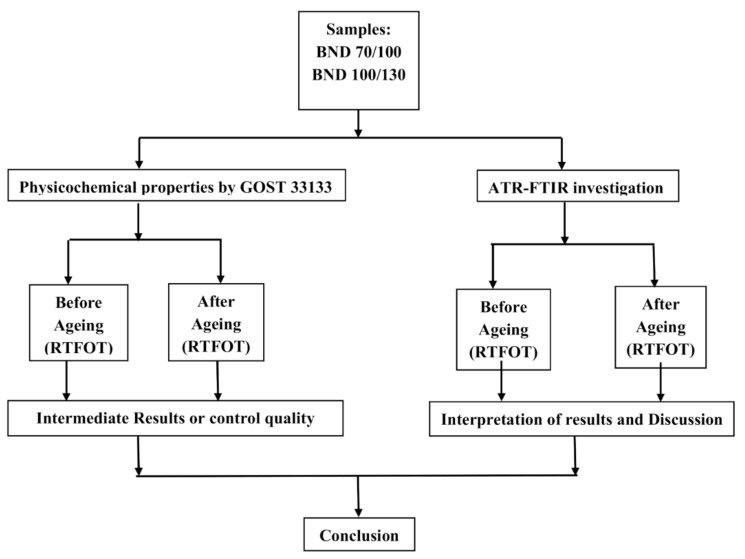
Flowchart of investigation design.

**Figure 2 materials-14-05870-f002:**
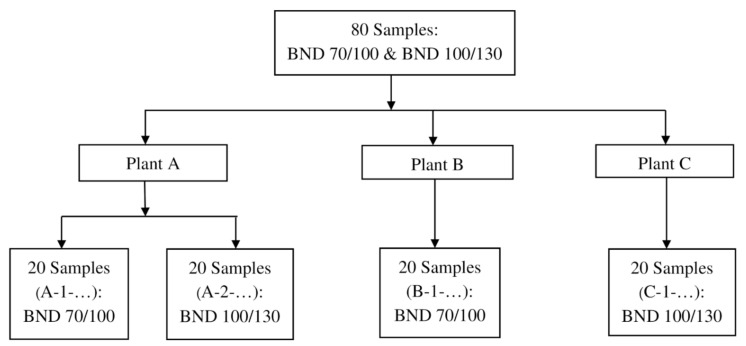
Flowchart of samples distribution in investigation.

**Figure 3 materials-14-05870-f003:**
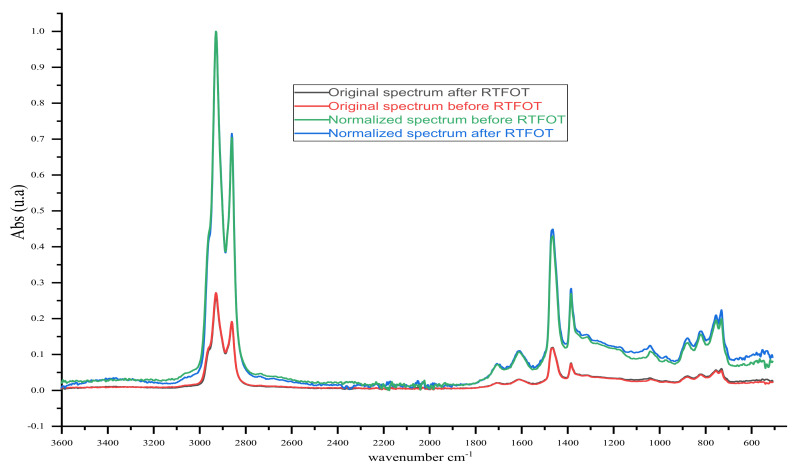
Example of original and normalised IR spectra of bitumen samples.

**Figure 4 materials-14-05870-f004:**
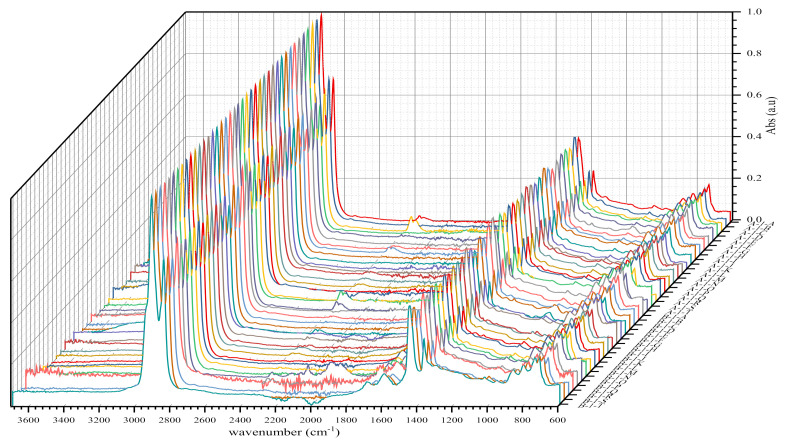
IR spectra of BND 70/100 from plants A and B before short-term ageing RTFOT.

**Figure 5 materials-14-05870-f005:**
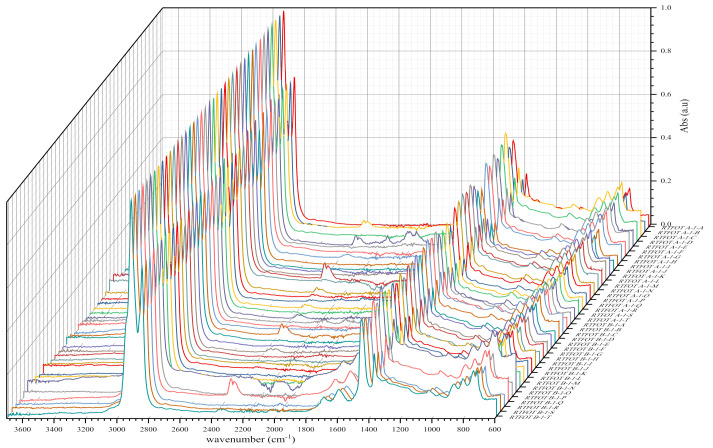
IR spectra of BND 70/100 from plants A and B after short-term ageing RTFOT.

**Figure 6 materials-14-05870-f006:**
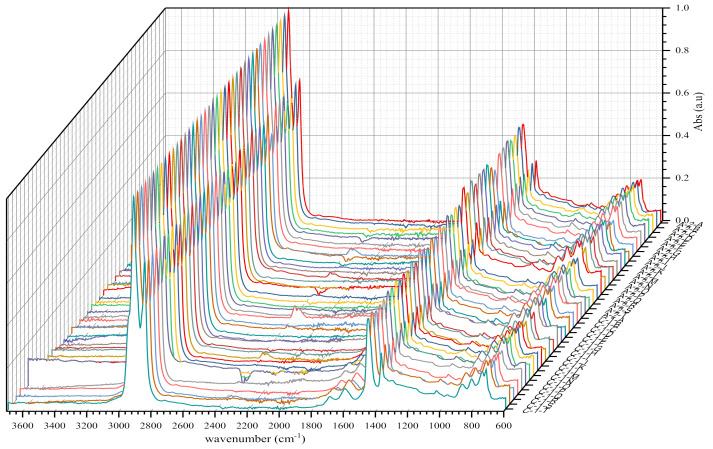
IR spectra of BND 100/130 from plants A and C before short-term ageing RTFOT.

**Figure 7 materials-14-05870-f007:**
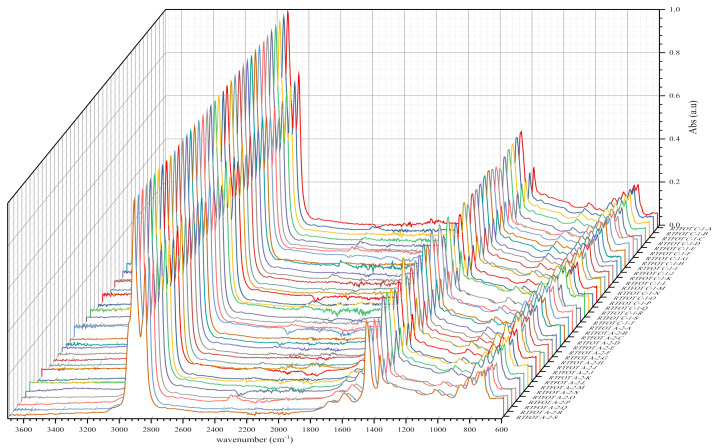
IR spectra of BND 100/130 from plants A and C after short-term ageing RTFOT.

**Figure 8 materials-14-05870-f008:**
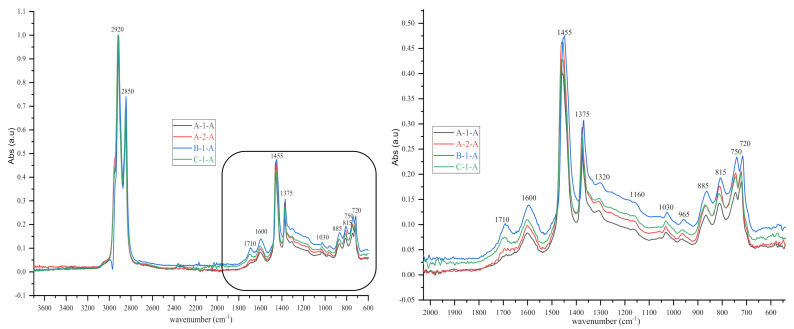
IR spectra (3700–600 cm^‒1^ and 2000–600 cm^‒1^) of some bituminous binders before short-term ageing (RTFOT).

**Figure 9 materials-14-05870-f009:**
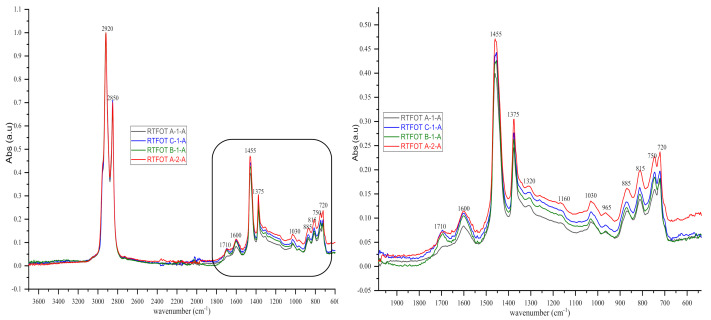
IR spectra (3700–600 cm^‒1^ and 2000–600 cm^‒1^) of some bituminous binders after short-term ageing RTFOT.

**Figure 10 materials-14-05870-f010:**
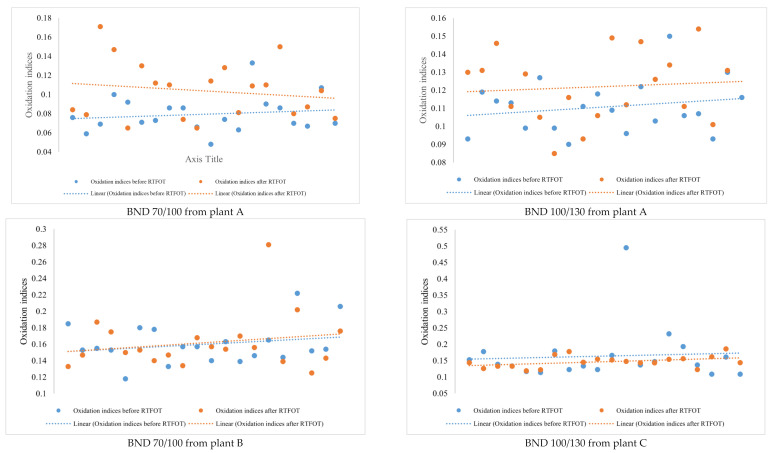
Comparison between the oxidation indices before and after RTFOT by plant and binder samples’ grades.

**Table 1 materials-14-05870-t001:** GOST 33133–2014, physicochemical properties of bitumen BND 70/100 and 100/130.

Designation	100/130	70/100	Test Methods
Indicators	Required Values		
Penetration 0.1 mm at 25 °C	101–130	71–100	GOST 33136
Softening point by ring and ball method, °C	≥45	≥47	GOST 33142
Ductility at 0 °C, cm	≥4.0	≥3.7	GOST 33138
Fraass breaking point, °C	≤−20	≤−18	GOST 33143
Rotational viscosity at 1.5 s^−1^ and 60 °C, Pa∙s before RTFOT	-		GOST 33137
Mass loss (%) after RTFOT ageing	≤0.7	≤0.6	GOST 33140
Difference in softening point after RTFOT, °C	≤7		GOST 33140/ GOST 33142
Rotational viscosity at 1.5 s^−1^ and 60 °C after RTFOT ageing, Pa’s	-		GOST 33140/ GOST 33137

**Table 2 materials-14-05870-t002:** The physicochemical properties of BND 70/100 from plant A.

Designation	BND 70/100 from Plant A									
Indicators	A-1-A	A-1-B	A-1-C	A-1-D	A-1-E	A-1-F	A-1-G	A-1-H	A-1-I	A-1-J
Penetration 0.1 mm at 25 °C	80	81	81	80	89	80	73	81	48	73
Softening point by ring and ball method, °C	48.8	47.6	47.2	48.4	48.4	48.2	49.6	47.4	48.2	50.6
Fraass breaking point, °C	−24	−23	−26	-	−27	-	-	-	−29.0	−22
Ductility at 0 °C, cm	3.50	crack	5.60	5.20	3.20	3.80	crack	4.70	3.90	3.20
Rotational viscosity at 1.5 s^−1^ and 60 °C, Pa∙s before RTFOT	217.37	143.96	149.63	201.74	116.33	156.28	200.30	181.97	234.21	189.40
Mass loss (%) after RTFOT ageing	0.04	0.04	0.07	0.05	0.04	0.04	0.02	0.07	0.05	0.04
Difference in softening point after RTFOT, °C	5.6	5.8	5.6	5.0	4.8	5.2	4.6	5.8	6.0	4.4
Rotational viscosity at 1.5s^−1^ and 60 °C after RTFOT ageing, Pa∙s	499.13	392.44	361.10	502.93	291.71	391.11	526.48	421.36	527.76	480.27
Indicators	A-1-K	A-1-L	A-1-M	A-1-N	A-1-O	A-1-P	A-1-Q	A-1-R	A-1-S	A-1-T
Penetration 0.1 mm at 25 °C	81	83	78	69	78	87	66	94	80	90
Softening point by ring and ball method, °C	50.2	51.6	49.8	48.8	47.0	47.0	49.6	47.6	48.8	48.0
Fraass breaking point, °C	−22	-	−18	-	−20	-	-	−27	-	-
Ductility at 0 °C, cm	3.80	3.10	3.90	3.90	crack	4.60	crack	3.80	crack	5.00
Rotational viscosity at 1.5 s^−1^ and 60 °C, Pa∙s before RTFOT	142.17	290.23	166.89	224.49	138.70	126.45	198.56	170.00	153.26	308.00
Mass loss (%) after RTFOT ageing	0.05	0.04	0.07	0.03	0.06	0.07	0.05	0.07	0.04	0.07
Difference in softening point after RTFOT, °C	4.2	6.0	5.2	4.8	5.2	5.0	4.0	5.0	4.6	3.8
Rotational viscosity at 1.5 s^−1^ and 60 °C after RTFOT ageing, Pa∙s	376.41	773.83	456.87	587.08	386.47	332.18	615.77	316.03	359.95	292.54

Red colour indicates that the results deviate from the norm.

**Table 3 materials-14-05870-t003:** The physicochemical properties of BND 100/130 from plant A.

Designation	BND 100/130 from Plant A									
Indicators	A-2-A	A-2-B	A-2-C	A-2-D	A-2-E	A-2-F	A-2-G	A-2-H	A-2-I	A-2-J
Penetration 0.1 mm at 25 °C	117.0	121.0	105.0	125.0	111.0	110.0	111.0	118.0	120.0	114.0
Softening point by ring and ball method, °C	45.3	45.4	45.0	44.8	44.6	45.0	45.4	45.6	46.0	45.4
Fraass breaking point, °C	−34	−35	−25	−26	−29	−30	−22	−22	−29	−20
Ductility at 0 °C, cm	4.60	5.10	4.80	4.30	5.10	5.70	4.80	4.40	4.80	4.70
Rotational viscosity at 1.5 s^−1^ and 60 °C, Pa∙s before RTFOT	120.30	114.50	127.90	104.68	182.60	176.90	173.20	144.88	109.00	139.70
Mass loss (%) after RTFOT ageing	0.60	0.80	0.80	0.60	0.60	0.60	0.60	0.80	0.90	0.80
Difference in softening point after RTFOT, °C	6.9	6.8	5.8	5.4	7.2	6.2	6.1	5.8	5.6	5.2
Rotational viscosity at 1.5s^−1^ and 60 °C after RTFOT ageing, Pa∙s	395.60	394.60	367.10	378.06	494.10	493.00	204.40	308.42	386.80	421.70
Indicators	A-2-K	A-2-L	A-2-M	A-2-N	A-2-O	A-2-P	A-2-Q	A-2-R	A-2-S	A-2-T
Penetration 0.1 mm at 25 °C	116.0	119.0	114.0	124.0	123.0	105.0	118.0	127.0	111.0	111.0
Softening point by ring and ball method, °C	45.6	45.2	45.4	43.6	44.4	45.4	46.0	45.2	45.6	45.2
Fraass breaking point, °C	−31	−29	−15	−27	−24	−26	−18	−28	−23	−27
Ductility at 0 °C, cm	4.90	5.00	4.40	4.50	5.10	4.90	4.80	5.00	5.00	4.50
Rotational viscosity at 1.5 s^−1^ and 60 °C, Pa∙s before RTFOT	168.70	150.60	141.61	137.50	136.00	137.20	154.14	119.80	132.75	131.90
Mass loss (%) after RTFOT ageing	0.80	0.80	0.80	0.80	0.04	0.80	0.70	0.90	0.80	0.10
Difference in softening point after RTFOT, °C	6.2	7.4	6.8	8.0	6.8	6.2	6.0	6.2	6.8	6.9
Rotational viscosity at 1.5 s^−1^ and 60 °C after RTFOT ageing, Pa∙s	390.90	422.30	422.14	369.50	395.10	413.60	429.62	391.23	424.27	414.00

Red colour indicates that the results deviate from the norm.

**Table 4 materials-14-05870-t004:** The physicochemical properties of BND 70/100 from plant B.

Designation	BND 70/100 from Plant B									
Indicators	B-1-A	B-1-B	B-1-C	B-1-D	B-1-E	B-1-F	B-1-G	B-1-H	B-1-I	B-1-J
Penetration 0.1 mm at 25 °C	84	85	81	81	75	75	75	81	81	76
Softening point by ring and ball method, °C	49.2	49.4	50.2	50.6	51.0	51.0	50.4	48.5	48.8	51.0
Fraass breaking point, °C	−34	-	-	-	-	-	-	-	-	-
Ductility at 0 °C, cm	4.40	4.40	4.30	4.10	4.00	3.90	4.10	4.20	3.80	4.20
Rotational viscosity at 1.5 s^−1^ and 60 °C, Pa∙s before RTFOT	351.00	357.89	412.29	390.04	423.37	420.38	418.19	346.42	350.19	447.35
Mass loss (%) after RTFOT ageing	0.20	0.10	0.10	0.11	0.20	0.14	0.14	0.12	0.14	0.13
Difference in softening point after RTFOT, °C	7.2	6.4	6.8	7.0	6.2	4.8	6.2	7.0	6.8	7.0
Rotational viscosity at 1.5 s^−1^ and 60 °C after RTFOT ageing, Pa∙s	890.70	970.34	1020.00	1041.40	925.60	971.24	1014.90	916.44	783.70	1020,6
Indicators	B-1-K	B-1-L	B-1-M	B-1-N	B-1-O	B-1-P	B-1-Q	B-1-R	B-1-S	B-1-T
Penetration 0.1 mm at 25 °C	83	77	75	80	81	76	78	87	75	74
Softening point by ring and ball method, °C	49.4	51.4	51.2	49.2	49.8	51.8	49.8	48.2	50.4	51.2
Fraass breaking point, °C	-	−22	−27	−33	−25	−33	−22	−26	−32	−28
Ductility at 0 °C, cm	4.0	4.1	3.4	3.7	4.0	4.2	3.8	3.7	3.5	3.7
Rotational viscosity at 1.5 s^−1^ and 60 °C, Pa∙s	399.81	431.48	400.30	385.00	353.60	431.99	407.24	275.98	456.80	461.78
Mass loss (%) after RTFOT ageing	0.13	0.14	0.10	0.20	0.20	0.10	0.13	0.12	0.10	0.13
Difference in softening point after RTFOT, °C	7.4	6,4	6.2	6.6	7.7	5.2	6.6	6.8	7.5	7.4
Rotational viscosity at 1.5 s^−1^ and 60 °C after RTFOT ageing, Pa∙s	887.56	1093.30	1051.00	925.30	929.00	951.40	1010.40	779.45	1104.10	1141.30

Red colour indicates that the results deviate from the norm.

**Table 5 materials-14-05870-t005:** The physicochemical properties of BND 100/130 from plant C.

Designation	BND 100/130 from Plant C									
Indicators	C-1-A	C-1-B	C-1-C	C-1-D	C-1-E	C-1-F	C-1-G	C-1-H	C-1-I	C-1-J
Penetration 0.1 mm at 25 °C	106	105	106	106	106	107	106	106	106	106
Softening point by ring and ball method, °C	45.8	45.8	45.6	46.0	45.2	46.6	45.6	45.4	45.6	45.6
Fraass breaking point, °C	−23	−22	−29	−21	−28	−28	−24	-	−29	−26
Ductility at 0 °C, cm	5.1	4.3	5.0	5.0	5.2	4.0	4.7	4.7	4.5	4.7
Rotational viscosity at 1.5 s^−1^ and 60 °C, Pa∙s before RTFOT	193.93	196.09	173.76	184.03	185.89	179.83	194.29	180.83	192.74	206.30
Mass loss (%) after RTFOT ageing	0.24	0.50	0.30	0.24	0.23	0.30	0.22	0.20	0.40	0.30
Difference in softening point after RTFOT, °C	5.40	7.60	5.40	4.40	6.20	5.40	5.40	5.80	6.40	5.60
Rotational viscosity at 1.5 s^−1^ and 60 °C after RTFOT ageing, Pa∙s	464.28	633.70	432.36	482.35	423.77	440.85	424.14	485.03	512.17	483.56
Indicators	C-1-K	C-1-L	C-1-M	C-1-N	C-1-O	C-1-P	C-1-Q	C-1-R	C-1-S	C-1-T
Penetration 0.1 mm at 25 °C	106	106	106	105	108	106	105	105	105	107
Softening point by ring and ball method, °C	45.2	45.4	45.4	45.8	45.4	45.2	45.4	45.8	45.6	46.2
Fraass breaking point, °C	−32	−31	−29	−26	-	−19	−25	−23	−19	−22
Ductility at 0 °C, cm	4.3	4.5	4.4	4.7	4.8	4.8	5.0	5.0	4.4	4.5
Rotational viscosity at 1.5 s^−1^ and 60 °C, Pa∙s before RTFOT	220.90	185.38	176.71	190.16	187.08	177.75	189.21	166.86	194.69	201.89
Mass loss (%) after RTFOT ageing	0.20	0.22	0.30	0.21	0.23	0.30	0.30	0.23	0.30	0.22
Difference in softening point after RTFOT, °C	6.20	6.60	6.80	6.00	5.60	5.80	6.20	5.60	6.00	6.20
Rotational viscosity at 1.5 s^−1^ and 60 °C after RTFOT ageing, Pa∙s	526.60	472.78	491.48	415.78	449.29	463.69	464.31	447.54	448.80	527.06

Red colour indicates that the results deviate from the norm.

**Table 6 materials-14-05870-t006:** Summary table of mean physicochemical properties BND 70/100 and 100/130.

Designation	BND 70/100						BND 100/130					
Indicators	Plant A (A-1-…)			Plant B (B-1-…)			Plant A (A-2-…)			Plant C (C-1-…)		
	Mean	SD	CV	Mean	SD	CV	Mean	SD	CV	Mean	SD	CV
Penetration 0.1 mm at 25 °C	80	9.83	12.30	79	3.91	4.95	116	6.32	5.45	106	0.76	0.72
Softening point by ring and ball method, °C	49	9.89	20.18	50	1.04	2.08	45.2	0.55	1.22	45.6	0.35	0.77
Fraass breaking point, °C	−24	3.46	14.41	−28	4.57	16.32	−26	5.06	19.46	−25	3.96	15.61
Ductility at 0 °C, cm	3	0.77	25.67	4	0.28	7.00	5	0.33	6.75	5	0.31	6.73
Rotational viscosity at 1.5 ^s‒1^ and 60 °C, Pa∙s before RTFOT	185.50	50.99	27.49	396.06	45.81	12.32	140.19	22.23	15.86	188.92	12.22	6.47
Mass loss (%) after RTFOT ageing	0.05	0.01	20.00	0.09	0.19	202.33	0.68	0.27	40.00	0.27	0.02	21.30
Difference in softening point after RTFOT, °C	5.03	0.65	12.92	6.34	0.74	11.67	6.42	1.55	25.22	5.9	0.66	11.17
Rotational viscosity at 1.5 s^‒1^ and 60 °C after RTFOT ageing, Pa∙s	444.57	122.12	27.47	920.36	98.11	10.65	395.82	60.70	15.34	451.26	50.82	10.70

Red colour indicates that the results deviate from the norm.

**Table 7 materials-14-05870-t007:** ATR-FTIR spectra’s data of bituminous binders.

Peak	Wavenumber (cm^−1^)	Assignation—Functional Group Components
1	2960 (±3)	C–H stretch asymmetric (alkyl groups)
2	2920 (±3)	C–H stretch asymmetric (alkyl groups)
3	2850 (±3)	C–H stretch symmetric (alkyl groups)
4	2360 (±3)	Molecules of carbon dioxide (CO_2_)
5	1710 (±3)	C=O stretch (carbonyls groups)
6	1600 (±3)	C=C stretch (aromatic structures)
7	1455 (±3)	C–H bend (aliphatic structures)
8	1375 (±3)	C–H bend (aliphatic structures)
9	1320 (±3)	C–O stretch (carbonyls groups)
10	1030 (±3)	S=O stretch (sulphoxide structures)
11	1160 (±3)	C–O stretch (Ether and oxy compound group)
12	885 (±3)	R4–C_6_H_2_ tetra substituted /R5–C_6_ H Penta substituted (aromatic structures)
13	815 (±3)	C–H bend1,4-disubstituted or 1,2,3,4-tetrasubstituted
14	750 (±3)	C=CH bend adjacent out of a plane (aromatic structures)
15	720 (±3)	C-H bend rocking (long chains)

**Table 8 materials-14-05870-t008:** Calculation of spectrometry indices of BND 70/100 from plant A before and after RTFOT.

	BND 70/100 from Plant A													
Spectrometry indices	A-1-A	RTFOT A-1-A	A-1-B	RTFOT A-1-B	A-1-C	RTFOT A-1-C	A-1-D	RTFOT A-1-D	A-1-E	RTFOT A-1-E	A-1-F		RTFOT A-1-F	
Aromaticity indices	0.449	0.468	0.457	0.465	0.442	0.543	0.481	0.525	0.471	0.392	0.441		0.498	
Oxidation indices	0.076	0.084	0.059	0.079	0.069	0.171	0.100	0.147	0.092	0.065	0.071		0.130	
Isomerisation indices	0.614	0.614	0.622	0.616	0.619	0.666	0.636	0.655	0.632	0.618	0.617		0.646	
Aliphaticity indices	5.247	5.068	5.316	5.127	5.421	4.172	5.039	4.383	4.977	5.739	5.500		4.615	
Sulphurisation indices	0.207	0.231	0.174	0.217	0.194	0.316	0.223	0.295	0.221	0.227	0.238		0.288	
Asphaltenes indices	0.361	0.365	0.348	0.359	0.355	0.398	0.369	0.386	0.365	0.358	0.352		0.376	
	BND 70/100 from plant A													
Spectrometry indices	A-1-G	RTFOT A-1-G	A-1-H	RTFOT A-1-H	A-1-I	RTFOT A-1-I	A-1-J	RTFOT A-1-J	A-1-K	RTFOT A-1-K	A-1-L	RTFOT A-1-L	A-1-M	RTFOT A-1-M
Aromaticity indices	0.454	0.439	0.475	0.505	0.461	0.371	0.450	0.392	0.437	0.461	0.461	0.507	0.445	0.448
Oxidation indices	0.073	0.112	0.086	0.110	0.086	0.074	0.066	0.065	0.048	0.114	0.074	0.128	0.063	0.081
Isomerisation indices	0.625	0.628	0.622	0.639	0.637	0.578	0.620	0.592	0.607	0.631	0.621	0.648	0.620	0.601
Aliphaticity indices	5.460	5.122	5.059	4.699	5.125	5.890	5.435	5.933	5.611	4.837	5.330	4.575	5.386	5.230
Sulphurisation indices	0.221	0.252	0.205	0.257	0.219	0.218	0.190	0.207	0.170	0.282	0.186	0.280	0.188	0.231
Asphaltenes indices	0.365	0.364	0.361	0.375	0.371	0.359	0.352	0.362	0.343	0.371	0.345	0.386	0.353	0.359
	BND 70/100 from plant A													
Spectrometry indices	A-1-N	RTFOT A-1-N	A-1-O	RTFOT A-1-O	A-1-P	RTFOTA-1-P	A-1-Q	RTFOT A-1-Q	A-1-R	RTFOT- A-1-R	A-1-S	RTFOT A-1-S	A-1-T	RTFOT A-1-T
Aromaticity indices	0.503	0.434	0.476	0.511	0.468	0.577	0.456	0.477	0.430	0.453	0.532	0.431	0.515	0.425
Oxidation indices	0.133	0.109	0.090	0.110	0.086	0.150	0.070	0.080	0.067	0.087	0.107	0.104	0.070	0.075
Isomerisation indices	0.645	0.614	0.634	0.642	0.630	0.655	0.619	0.623	0.623	0.627	0.642	0.599	0.611	0.611
Aliphaticity indices	4.559	5.161	4.986	4.609	5.142	4.215	5.293	5.085	5.470	5.187	4.357	5.122	5.069	5.426
Sulphurisation indices	0.274	0.270	0.214	0.256	0.225	0.283	0.214	0.223	0.205	0.239	0.238	0.250	0.183	0.226
Asphaltenes indices	0.381	0.368	0.362	0.369	0.366	0.387	0.357	0.361	0.358	0.370	0.365	0.370	0.356	0.359

Red colour indicates that the results deviate from the norm.

**Table 9 materials-14-05870-t009:** Calculation of spectrometry indices of BND 100/130 from plant A before and after RTFOT.

	BND 100/130 from Plant A													
Spectrometry indices	A-2-A	RTFOT A-2-A	A-2-B	RTFOT A-2-B	A-2-C	RTFOT A-2-C	A-2-D	RTFOT A-2-D	A-2-E	RTFOT A-2-E	A-2-F	RTFOT A-2-F	A-2-G	RTFOT A-2-G
Aromaticity indices	0.479	0.484	0.513	0.487	0.483	0.546	0.506	0.495	0.492	0.492	0.506	0.475	0.486	0.438
Oxidation indices	0.093	0.130	0.119	0.131	0.114	0.146	0.113	0.111	0.099	0.129	0.127	0.105	0.099	0.085
Isomerisation indices	0.633	0.647	0.642	0.644	0.650	0.663	0.646	0.640	0.642	0.650	0.659	0.636	0.640	0.630
Aliphaticity indices	5.075	4.643	4.676	4.686	4.903	4.348	4.682	4.881	4.863	4.693	4.601	4.907	4.902	5.195
Sulphurisation indices	0.207	0.287	0.239	0.283	0.259	0.285	0.236	0.253	0.249	0.268	0.272	0.254	0.224	0.248
Asphaltenes indices	0.346	0.366	0.361	0.367	0.353	0.368	0.354	0.357	0.341	0.358	0.355	0.345	0.341	0.341
	BND 100/130 from plant A													
Spectrometry indices	A-2-H	RTFOT A-2-H	A-2-I	RTFOT A-2-I	A-2-J	RTFOT A-2-J	A-2-K	RTFOT A-2-K	A-2-L	RTFOT A-2-L	A-2-M	RTFOT A-2-M	A-2-N	RTFOT A-2-N
Aromaticity indices	0.458	0.483	0.511	0.449	0.508	0.441	0.499	0.522	0.482	0.491	0.513	0.467	0.492	0.496
Oxidation indices	0.090	0.116	0.111	0.093	0.118	0.106	0.109	0.149	0.096	0.112	0.122	0.147	0.103	0.126
Isomerisation indices	0.629	0.637	0.620	0.623	0.641	0.645	0.637	0.645	0.651	0.633	0.645	0.642	0.636	0.643
Aliphaticity indices	5.466	4.875	4.882	5.077	4.704	5.143	4.824	4.376	5.015	4.840	4.648	4.761	4.966	4.680
Sulphurisation indices	0.220	0.251	0.178	0.244	0.238	0.268	0.236	0.284	0.217	0.258	0.250	0.282	0.233	0.260
Asphaltenes indices	0.342	0.356	0.338	0.356	0.353	0.349	0.353	0.363	0.343	0.356	0.361	0.366	0.344	0.358
	BND 100/130 from plant A													
Spectrometry indices	A-2-O	RTFOT A-2-O	A-2-P	RTFOT A-2-P	A-2-Q	RTFOT A-2-Q	A-2-R	RTFOT A-2-R	A-2-S		RTFOT A-2-S		A-2-T	
Aromaticity indices	0.531	0.518	0.499	0.452	0.494	0.504	0.484	0.411	0.519		0.498		0.500	
Oxidation indices	0.150	0.134	0.106	0.111	0.107	0.154	0.093	0.101	0.130		0.131		0.116	
Isomerisation indices	0.673	0.651	0.635	0.614	0.637	0.633	0.630	0.609	0.647		0.639		0.652	
Aliphaticity indices	4.394	4.538	4.894	4.965	4.909	4.434	5.101	5.131	4.611		4.608		4.796	
Sulphurisation indices	0.301	0.277	0.216	0.233	0.225	0.282	0.206	0.250	0.251		0.280		0.265	
Asphaltenes indices	0.367	0.359	0.352	0.358	0.350	0.376	0.344	0.356	0.362		0.366		0.354	

Red colour indicates that the results deviate from the norm.

**Table 10 materials-14-05870-t010:** Calculation of spectrometry indices of BND 70/100 from plant B before and after RTFOT.

	BND 70/100 from Plant B													
Spectrometry indices	B-1-A	RTFOT B-1-A	B-1-B	RTFOT B-1-B	B-1-C	RTFOT B-1-C	B-1-D	RTFOT B-1-D	B-1-E	RTFOT B-1-E	B-1-F	RTFOT B-1-F	B-1-G	RTFOT B-1-G
Aromaticity indices	0.586	0.575	0.554	0.572	0.549	0.587	0.548	0.568	0.508	0.539	0.588	0.556	0.592	0.599
Oxidation indices	0.185	0.133	0.153	0.147	0.155	0.187	0.153	0.175	0.118	0.150	0.180	0.153	0.178	0.140
Isomerisation indices	0.655	0.622	0.647	0.631	0.638	0.654	0.641	0.642	0.631	0.628	0.650	0.639	0.655	0.622
Aliphaticity indices	3.922	4.249	4.248	4.275	4.267	3.979	4.293	4.123	4.760	4.397	3.995	4.290	3.910	4.239
Sulphurisation indices	0.265	0.233	0.243	0.246	0.252	0.299	0.280	0.272	0.266	0.236	0.264	0.252	0.279	0.202
Asphaltenes indices	0.379	0.359	0.363	0.361	0.374	0.387	0.363	0.380	0.356	0.359	0.375	0.369	0.381	0.365
	BND 70/100 from plant B													
Spectrometry indices	B-1-O	RTFOT B-1-O	B-1-P	RTFOTB-1-P	B-1-Q	RTFOT B-1-Q	B-1-R	RTFOT- B-1-R	B-1-S	RTFOT B-1-S		B-1-T		RTFOTB-1-T
Aromaticity indices	0.519	0.655	0.537	0.538	0.625	0.604	0.569	0.517	0.552	0.521		0.648		0.596
Oxidation indices	0.165	0.281	0.144	0.139	0.222	0.202	0.152	0.125	0.154	0.143		0.206		0.176
Isomerisation indices	0.644	0.698	0.637	0.638	0.678	0.660	0.639	0.623	0.645	0.632		0.667		0.649
Aliphaticity indices	4.323	3.378	4.368	4.405	3.759	3.791	4.269	4.583	4.243	4.450		3.652		3.979
Sulphurisation indices	0.261	0.375	0.229	0.233	0.259	0.312	0.208	0.223	0.260	0.250		0.283		0.276
Asphaltenes indices	0.375	0.405	0.370	0.364	0.366	0.391	0.364	0.358	0.370	0.372		0.394		0.381

Red colour indicates that the results deviate from the norm.

**Table 11 materials-14-05870-t011:** Calculation of spectrometry indices of BND 100/130 from plant C before and after RTFOT.

	BND 100/130 from Plant C													
Spectrometry indices	C-1-A	RTFOT C-1-A	C-1-B	RTFOT C-1-B	C-1-C	RTFOT C-1-C	C-1-D	RTFOT C-1-D	C-1-E	RTFOT C-1-E	C-1-F	RTFOT C-1-F	C-1-G	RTFOT C-1-G
Aromaticity indices	0.544	0.558	0.559	0.512	0.531	0.518	0.520	0.535	0.522	0.519	0.570	0.512	0.565	0.535
Oxidation indices	0.153	0.144	0.178	0.126	0.139	0.133	0.134	0.133	0.117	0.119	0.114	0.122	0.180	0.169
Isomerisation indices	0.635	0.633	0.647	0.627	0.631	0.630	0.630	0.625	0.618	0.615	0.629	0.617	0.647	0.630
Aliphaticity indices	4.319	4.302	4.151	4.586	4.434	4.516	4.490	4.458	4.617	4.639	4.091	4.784	4.065	4.196
Sulphurisation indices	0.250	0.258	0.275	0.249	0.247	0.259	0.240	0.250	0.217	0.217	0.232	0.237	0.282	0.294
Asphaltenes indices	0.377	0.368	0.383	0.368	0.373	0.368	0.366	0.370	0.360	0.359	0.355	0.368	0.385	0.385
	BND 100/130 from plant C													
Spectrometry indices	C-1-H	RTFOT C-1-H	C-1-I	RTFOT C-1-I	C-1-J	RTFOT C-1-J	C-1-K	RTFOT C-1-K	C-1-L	RTFOT C-1-L	C-1-M	RTFOT C-1-M	C-1-N	
Aromaticity indices	0.518	0.588	0.502	0.533	0.533	0.593	0.553	0.549	0.876	0.552	0.493	0.538	0.540	
Oxidation indices	0.123	0.178	0.134	0.145	0.123	0.155	0.166	0.152	0.495	0.148	0.137	0.143	0.147	
Isomerisation indices	0.618	0.645	0.632	0.630	0.622	0.644	0.643	0.630	1.146	0.633	0.637	0.632	0.635	
Aliphaticity indices	4.598	4.027	4.566	4.356	4.505	4.045	4.210	4.311	3.325	4.278	4.592	4.379	4.338	
Sulphurisation indices	0.216	0.284	0.243	0.269	0.222	0.267	0.272	0.257	0.657	0.265	0.281	0.255	0.254	
Asphaltenes indices	0.362	0.387	0.367	0.376	0.370	0.367	0.384	0.372	0.536	0.379	0.375	0.370	0.379	
	BND 100/130 from plant C													
Spectrometry indices	RTFOT C-1-N	C-1-O	RTFOTC-1-O	C-1-P	RTFOT C-1-P	C-1-Q	RTFOT C-1-Q	C-1-R	RTFOT C-1-R	C-1-S	RTFOTC-1-S	C-1-T	RTFOTC-1-T	
Aromaticity indices	0.545	0.646	0.540	0.670	0.562	0.535	0.489	0.493	0.572	0.551	0.561	0.518	0.558	
Oxidation indices	0.143	0.232	0.154	0.193	0.156	0.137	0.123	0.109	0.162	0.161	0.186	0.109	0.144	
Isomerisation indices	0.627	0.664	0.641	0.632	0.644	0.627	0.608	0.615	0.635	0.647	0.659	0.617	0.623	
Aliphaticity indices	4.423	3.584	4.308	3.662	4.327	4.440	4.682	4.834	4.220	4.232	4.091	4.624	4.415	
Sulphurisation indices	0.247	0.312	0.274	0.293	0.257	0.228	0.244	0.204	0.274	0.276	0.308	0.205	0.241	
Asphaltenes indices	0.377	0.396	0.378	0.415	0.377	0.370	0.365	0.357	0.392	0.387	0.374	0.355	0.377	

Red colour indicates that the results deviate from the norm.

**Table 12 materials-14-05870-t012:** The mean of spectrometry indices per plant with their standard deviation (SD) and their variance coefficient (CV).

Spectrometry Indices	BND 70/100 from Plant A						BND 100/130 from Plant A					
	Mean Initial	SD	CV	Mean RTFOT	SD	CV	Mean Initial	SD	CV	Mean RTFOT	SD	CV
Aromaticity indices	0.465	0.026	5.7	0.466	0.053	11.4	0.498	0.017	3.3	0.482	0.033	6.8
Oxidation indices	0.079	0.019	24.0	0.104	0.030	29.0	0.111	0.015	13.3	0.122	0.020	16.1
Isomerisation indices	0.625	0.010	1.6	0.625	0.023	3.7	0.642	0.012	1.8	0.638	0.013	2.0
Aliphaticity indices	5.189	0.313	6.0	5.010	0.505	10.1	4.846	0.229	4.7	4.778	0.257	5.4
Sulphurisation indices	0.209	0.025	11.9	0.252	0.031	12.1	0.236	0.027	11.5	0.266	0.017	6.3
Asphaltenes indices	0.359	0.009	2.6	0.370	0.011	3.1	0.351	0.008	2.3	0.359	0.008	2.3
Spectrometry indices	BND 70/100 from plant B						BND 100/130 from plant C					
	Mean Initial	SD	CV	Mean RTFOT	SD	CV	Mean Initial	SD	CV	Mean RTFOT	SD	CV
Aromaticity indices	0.561	0.034	6.0	0.568	0.032	5.6	0.562	0.086	15.3	0.543	0.026	4.8
Oxidation indices	0.160	0.024	15.3	0.162	0.034	21.0	0.164	0.084	51.2	0.147	0.018	12.4
Isomerisation indices	0.645	0.012	1.9	0.639	0.017	2.7	0.659	0.115	17.5	0.631	0.012	1.8
Aliphaticity indices	4.209	0.262	6.2	4.188	0.268	6.4	4.284	0.387	9.0	4.367	0.205	4.7
Sulphurisation indices	0.248	0.024	9.6	0.259	0.037	14.3	0.270	0.096	35.5	0.260	0.021	7.9
Asphaltenes indices	0.370	0.009	2.5	0.372	0.012	3.3	0.383	0.039	10.2	0.374	0.008	2.1

The red colour indicates that the results deviate from the norm. Mean Initial is the mean of spectrometry indices before RTFOT. Mean RTFOT is the mean of spectrometry indices after RTFOT.

**Table 13 materials-14-05870-t013:** Summarising the mean physicochemical properties of the bituminous binders studied.

Designation	BND 70/100		BND 100/130	
Indicators	Plant A	Plant B	Plant A	Plant C
	Mean	Mean	Mean	Mean
Penetration 0.1 mm at 25 °C	80	79	116	106
Softening point by ring and ball method, °C	49	50	45.2	45.6
Fraass breaking point, °C	−24	−28	−26	−25
Ductility at 0 °C, cm	3	4	5	5
Rotational viscosity at 1.5 s^v‒1^ and 60 °C, Pa∙s before RTFOT	185.50	396.06	140.19	188.92
Mass loss (%) after RTFOT ageing	0.05	0.09	0.68	0.27
Difference in softening point after RTFOT, °C	5.03	6.34	6.42	5.9
Rotational viscosity at 1.5 s^‒1^ and 60 °C after RTFOT ageing, Pa∙s	444.57	920.36	395.82	451.26

**Table 14 materials-14-05870-t014:** Summary of Fourier-transform infrared spectroscopy investigation.

Designation	BND 70/100 from Plant A		BND 100/130 from Plant A		BND 70/100 from Plant B		BND 100/130 from Plant C	
Visual Discrepancy between Samples	Almost no Band at 1710 cm^−1^(C=O)		The Presence of Band at 1710 cm^−1^(C=O)					
Discrepancy between samples based on mean values of spectrometry indices	Oxidation indices							
	Mean Initial	Mean RTFOT	Mean Initial	Mean RTFOT	Mean Initial	Mean RTFOT	Mean Initial	Mean RTFOT
	0.079	0.104	0.111	0.122	0.160	0.162	0.164	0.147
	CV = 24.0	CV = 29.0	CV = 13.3	CV = 16.1	CV = 15.3	CV = 21.0	CV = 51.2	CV = 12.4
	The oxidation degree is relatively low, even if we consider before and after RTFOT; the mean is 0.092.		The oxidation degree is relatively low, even if we consider before and after RTFOT; the mean is 0.117.		The oxidation degree is relatively high but does not change even after RTFOT; the mean is 0.161.		The oxidation degree is relatively high before RTFOT, but after RTFOT, we observe a decrease of oxidation indices; the mean is 0.156.	
	Aromaticity indices							
	Mean Initial	Mean RTFOT	Mean Initial	Mean RTFOT	Mean Initial	Mean RTFOT	Mean Initial	Mean RTFOT
	0.465	0.466	0.498	0.481	0.561	0.568	0.562	0.543
	CV = 5.7	CV = 11.4	CV = 3.3	CV = 6.8	CV = 6.0	CV = 5.6	CV = 15.3	CV = 4.8
	Aliphaticity indices							
	Mean Initial	Mean RTFOT	Mean Initial	Mean RTFOT	Mean Initial	Mean RTFOT	Mean Initial	Mean RTFOT
	5.189	5.010	4.845	4.778	4.209	4.188	4.284	4.367
	CV = 6.0	CV = 10.1	CV = 4.7	CV = 5.4	CV = 6.2	CV = 6.4	CV = 9.0	CV = 4.7

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
