# Peer review of "An Attempt to Track Two Grades of Road Bitumen from Different Plants Using Fourier Transform Infrared Spectroscopy"

_materials, 2021, doi:10.3390/ma14195870_

Round 1
Reviewer 1 Report
In general, the manuscript should be more concisely written, including the current state-of-the-art. The practical potential of this work is also limited, as the data should be analyzed statistically. Please consider the following comments to improve the paper's quality.
- Lines 71-76: Important studies that have introduced FTIR spectroscopy in bituminous binders are missing. In Introduction, it is highly recommended to consider the works from Western Research Institute and Petersen.
- Lines 84-90: The aging in bituminous binders has been a topic well studied in the last decades. Indeed, FTIR spectroscopy has contributed a lot to establishing the theoretical background to explain and interpret aging. Recently, a lot of effort has been spent by various research groups, such as TU Delft and Apostolidis, on resolving the degradation of binders due to oxidative aging by implementing anti-aging technologies, see epoxy-based compounds. It is recommended to include such studies in this part of the manuscript to support the community with the current state-of-the-art.
- As the acronym of Fourier Transform Infrared spectroscopy was introduced, please keep using FTIR. For the spectra, the authors could say infrared (IR) spectra instead of just spectra. Please resolve accordingly in the whole manuscript.
- It is recommended to put some pages horizontally to include tables without continuation.
- In sub-section 2.1, please provide the standards and specifications followed to age the samples. Similarly, please mention the spec of FTIR spectroscopy in sub-section 2.2.
- In sub-section 2.2.2, please place the description after each equation. Also, did you mean carbonyls index instead of oxidation?
- Did the authors find any correlation between the chemistry analyzed in FTIR and the physical properties provided in Tables 2-6? Also, did the authors need to provide Tables 2-5 as Table 6 gives the main values of binders?
- It is believed that Table 7 should be placed after sub-section 2.2.2 in a new sub-section mainly created to discuss the main functionalities of bituminous binders and those used for fingerprinting.
- In tables, why are some values red-colored? What is also the meaning of BF?
As mentioned above, there are significant language issues that need to be resolved by the authors. Here, the reviewer focused on the abstract section providing some comments and suggestions to improve the quality. It is expected the authors follow a similar style over proofreading the manuscript.
- ‘This study aims to evaluate the possibility of using Fourier Transform Infrared (FTIR) spectroscopy to track binders produced by three different plants; plants A, B, and C. The work included the quality assessment of bituminous binders graded as 70/100 and 100/130 according to GOST 33133 (Russian interstate standard) and chemical analyses using FTIR spectroscopy. FTIR analyses were also conducted after short-term aging in a Rolling Thin Film Oven Test (RTFOT). All infrared spectra were normalized to ensure the reliability of results, and the standard deviation and variance coefficient were included. The ultimate goal of the present study was to track the origin and the aging extent of studied bituminous binders.
Author Response
Dear Reviewer,
Thank you very much for your valuable comments and attention to our manuscript. Your feedback helps us improve this manuscript. We hope that at best we will respond to your recommendations.
Your comments have been taken into account, see the manuscript and cover letter.

Reviewer 2 Report
General:
Abstract – you should include what is the purpose of your studies? What is the motivation of your research and what are your main findings. I suggest to reorder the abstract and to mention in it the reasons of studies, which you presented in the Introduction.
The form of presenting results is too detailed and not clear. You repeat numerous figures and tables with very close shapes or values. It is difficult to find any relationships or trends basing on currently figures and tables. I recommend to simplified and make the form of results presentation more attractive.
It is not clear how you assume the reference FTiR spectra and how the acceptable level of standard deviation is set.
The paper seems to be a report from laboratory test conducted for Gazprom. It is difficult to find the novel aspects in presented studies or to find a practical findings for a wider group of road engineers.
Detailed comments:
Section 2.1. It is not clear how did you collect the samples? Give more details.
Table 1 – check the value required for softening point (7 deg C), there is a mistake, or you should rename the designation. Softening point after RTFOT is higher than for origin bitumen, and for origin the required value is higher than 45 deg C. Analogously in tables from 2 to 5.
Figure 3 – it is not clear how did you obtain the IR spectra? Which samples did you used? I guess that there were origin samples collected directly from the refinery but it is not clear.
Figure 3 – How did you consider the variability of production. You refer to studies of Hofko et al. but you should include some more details to make possible checking the charts.
Tables 2 – 5. What is the purpose of repeating similar tables? Similarly what is a sense of presenting very similar charts in figures 5 – 11? There are not practical thus I recommend to authors to propose more clear form of presenting results of their laboratory test.
You identified on the basis of FTiR that the difference are related with oxidation process. Does this observation relate to the changes in physicochemical properties (penetration etc.?)
Figure 12 – give the description of y-axle.
I recommend to shit results from Tables 8 – 11 into a new figure. It is not clear for me if values in table 8-11 are the same as in Figure 12?
Author Response

(The authors gave the same response as above.)

Round 2
Reviewer 2 Report
The paper presents much better after the review. However following aspects should be considered:
- I think that still there is too little attention paid on process of sample collection. FTIR is a very sensitive method and in a case of contamination of the sample or in a case of inappropriate collection of the bitumen the result of FTIR test can be unrepresentative for a given part of the product. In previous remark I asked to describe in more details how did you collected the specimens of the bitumen. I did not mean the number of samples or combinations. I meant that you should describe the amount of bitumen per each of 80 specimens, pace of collection (directly from plant installation?), if you blended bitumens from different parts of production, process of specimen storage etc. Please include more detailed description about appropriate data collection and how to minimalize the impact of inhomogeneity of production on FTIR results. Pleas also include some comments about that in conclusion section.
- In conclusions you wrote: “it is recommended to introduce unique chemical markers (tracers) to achieve effective traceability and identify Gazpromneft-Bituminous Materials products” Could you present those chemical marker on the base of results given in this studies. In table 14 you discuss about the discrepancy, I think that table 14 is also a good place to present the reference values (or range of values) of the following indexes: oxidation, aromaticity, aliphaticity, before and after RTFOT. You can also present table 14 graphically, it will be much more attractive.
- Please also check language and spelling mistakes
